# Incidence of diabetes following COVID-19 vaccination and SARS-CoV-2 infection in Hong Kong: A population-based cohort study

Xi Xiong[1‡], David Tak Wai Lui[2‡], Matthew Shing Hin Chung[1], Ivan Chi Ho Au[1], Francisco Tsz Tsun Lai[1,3], Eric Yuk Fai Wan[1,3,4], Celine Sze Ling Chui[3,5,6], Xue Li[1,2,3], Franco Wing Tak Cheng[1], Ching-Lung Cheung[1,3], Esther Wai Yin Chan[1,3,7,8], Chi Ho Lee[2], Yu Cho Woo[2], Kathryn Choon Beng Tan[2], Carlos King Ho Wong[1,3,4]*, Ian Chi Kei Wong[1,3,9,10]*

1 Centre for Safe Medication Practice and Research, Department of Pharmacology and Pharmacy, Li Ka Shing Faculty of Medicine, The University of Hong Kong, Hong Kong SAR, China, 2 Department of Medicine, School of Clinical Medicine, Li Ka Shing Faculty of Medicine, The University of Hong Kong, Hong Kong SAR, China, 3 Laboratory of Data Discovery for Health (D²4H), Hong Kong Science and Technology Park, Hong Kong SAR, China, 4 Department of Family Medicine and Primary Care, School of Clinical Medicine, Li Ka Shing Faculty of Medicine, The University of Hong Kong, Hong Kong SAR, China, 5 School of Nursing, Li Ka Shing Faculty of Medicine, The University of Hong Kong, Hong Kong SAR, China, 6 School of Public Health, Li Ka Shing Faculty of Medicine, The University of Hong Kong, Hong Kong SAR, China, 7 Department of Pharmacy, The University of Hong Kong-Shenzhen Hospital, Shenzhen, China, 8 The University of Hong Kong Shenzhen Institute of Research and Innovation, Shenzhen, China, 9 Aston Pharmacy School, Aston University, Birmingham, United Kingdom, 10 Research Department of Practice and Policy, School of Pharmacy, University College London, London, United Kingdom

‡ These authors share first authorship on this work.
* carlosho@hku.hk (CKHW); wongick@hku.hk (ICKW)

## Abstract

### Background

The risk of incident diabetes following Coronavirus Disease 2019 (COVID-19) vaccination remains to be elucidated. Also, it is unclear whether the risk of incident diabetes after Severe Acute Respiratory Syndrome Coronavirus 2 (SARS-CoV-2) infection is modified by vaccination status or differs by SARS-CoV-2 variants. We evaluated the incidence of diabetes following mRNA (BNT162b2), inactivated (CoronaVac) COVID-19 vaccines, and after SARS-CoV-2 infection.

### Methods and findings

In this population-based cohort study, individuals without known diabetes were identified from an electronic health database in Hong Kong. The first cohort included people who received ≥1 dose of COVID-19 vaccine and those who did not receive any COVID-19 vaccines up to September 2021. The second cohort consisted of confirmed COVID-19 patients and people who were never infected up to March 2022. Both cohorts were followed until August 15, 2022. A total of 325,715 COVID-19 vaccine recipients (CoronaVac: 167,337; BNT162b2: 158,378) and 145,199 COVID-19 patients were 1:1 matched to their respective controls using propensity score for various baseline characteristics. We also adjusted for

the study of LONG COVID and COVID-19 vaccine safety monitoring. The vaccination and infection record data are owned by the Department of Health. Clinical records are owned by Hospital Authority. Vaccination and infection records were linked to clinical records on de-identified patients of the Hospital Authority separately. Data cannot be shared publicly because authors are bound by ethical, legal and contractual conditions imposed by both Department of Health and the Hospital Authority, and are not allowed to use the data for any other purposes or divulge the data to any third parties. Following approvals from the Institutional Review Board, data requests were submitted and assessed by both Department of Health and Hospital Authority prior to data release for use by specified research delegates only. For further information regarding the data request and approval process, please see the website of Hospital Authority for provision of data for research: https://www3.ha.org.hk/data/Provision/Submission. Hospital Authority data access inquiries can be directed to hacpaaedr@ha.org.hk.

**Funding:** This work was supported by a research grant from the Health Bureau; HMRF Research on COVID-19, The Government of the Hong Kong Special Administrative Region (principal investigator, ICKW; reference no. COVID19F01); a research grant from the Health Bureau; HMRF Research on COVID-19, The Government of the Hong Kong Special Administrative Region (principal investigator [work package 2], EWYC; reference no. COVID1903011); Collaborative Research Fund, University Grants Committee, HKSAR Government (principal investigator, ICKW; reference no. C7154-20GF). ICKW and FTTL are partially supported by the Laboratory of Data Discovery for Health (D24H) funded by the by AIR@InnoHK administered by Innovation and Technology Commission. The funders had no role in study design, data collection and analysis, decision to publish, or preparation of the manuscript.

**Competing interests:** I have read the journal's policy and the authors of this manuscript have the following competing interests: CKHW reports receipt of research funding from the EuroQoL Group Research Foundation, the Hong Kong Research Grants Council, the Hong Kong Health and Medical Research Fund; AstraZeneca and Boehringer Ingelheim, unrelated to this work. FTTL has been supported by the RGC Postdoctoral Fellowship under the Hong Kong Research Grants Council. EYFW has received research grants from the Food and Health Bureau of the Government of the Hong Kong SAR, and the Hong Kong Research

previous SARS-CoV-2 infection when estimating the conditional probability of receiving vaccinations, and vaccination status when estimating the conditional probability of contracting SARS-CoV-2 infection. Hazard ratios (HRs) and 95% confidence intervals (CIs) for incident diabetes were estimated using Cox regression models.

In the first cohort, we identified 5,760 and 4,411 diabetes cases after receiving CoronaVac and BNT162b2 vaccines, respectively. Upon a median follow-up of 384 to 386 days, there was no evidence of increased risks of incident diabetes following CoronaVac or BNT162b2 vaccination (CoronaVac: 9.08 versus 9.10 per 100,000 person-days, HR = 0.998 [95% CI 0.962 to 1.035]; BNT162b2: 7.41 versus 8.58, HR = 0.862 [0.828 to 0.897]), regardless of diabetes type. In the second cohort, we observed 2,109 cases of diabetes following SARS-CoV-2 infection. Upon a median follow-up of 164 days, SARS-CoV-2 infection was associated with significantly higher risk of incident diabetes (9.04 versus 7.38, HR = 1.225 [1.150 to 1.305])—mainly type 2 diabetes—regardless of predominant circulating variants, albeit lower with Omicron variants (p for interaction = 0.009). The number needed to harm at 6 months was 406 for 1 additional diabetes case. Subgroup analysis revealed no evidence of increased risk of incident diabetes among fully vaccinated COVID-19 survivors. Main limitations of our study included possible misclassification bias as type 1 diabetes was identified through diagnostic coding and possible residual confounders due to its observational nature.

## Conclusions

There was no evidence of increased risks of incident diabetes following COVID-19 vaccination. The risk of incident diabetes increased following SARS-CoV-2 infection, mainly type 2 diabetes. The excess risk was lower, but still statistically significant, for Omicron variants. Fully vaccinated individuals might be protected from risks of incident diabetes following SARS-CoV-2 infection.

## Author summary

### Why was this study done?

- There have been an increasing number of cases of type 1 diabetes reported following Coronavirus Disease 2019 (COVID-19) vaccinations.

- The relationship between receiving COVID-19 vaccines and incident diabetes has not been examined in population-based studies.

- Several nationwide cohorts reported higher risks of incident diabetes following Severe Acute Respiratory Syndrome Coronavirus 2 (SARS-CoV-2) infection.

- The risk of incident diabetes following infection by SARS-CoV-2 Omicron variants may differ from that following infection by earlier variants. It is also uncertain how vaccination status may influence the risk.

Grants Council, outside the submitted work. CSLC has received grants from the Food and Health Bureau of the Hong Kong Government, Hong Kong Research Grant Council, Hong Kong Innovation and Technology Commission, Pfizer, IQVIA, and Amgen; personal fee from Primevigilance Ltd.; outside the submitted work. XL has received research grants from the Food and Health Bureau of the Government of the Hong Kong SAR, research and educational grants from Janssen and Pfizer; internal funding from University of Hong Kong; consultancy fee from Merck Sharp & Dohme, unrelated to this work. EWYC reports honorarium from Hospital Authority, grants from Research Grants Council (RGC, Hong Kong), grants from Research Fund Secretariat of the Food and Health Bureau, grants from National Natural Science Fund of China, grants from Wellcome Trust, grants from Bayer, grants from Bristol-Myers Squibb, grants from Pfizer, grants from Janssen, grants from Amgen, grants from Takeda, grants from Narcotics Division of the Security Bureau of HKSAR, outside the submitted work. ICKW reports research funding outside the submitted work from Amgen, Bristol-Myers Squibb, Pfizer, Janssen, Bayer, GSK, Novartis, the Hong Kong RGC, and the Hong Kong Health and Medical Research Fund, National Institute for Health Research in England, European Commission, National Health and Medical Research Council in Australia, and also received speaker fees from Janssen and Medice in the previous 3 years. All other authors declare no competing interests.

**Abbreviations:** ACE2, angiotensin-converting enzyme 2; ASIA, autoimmune/inflammatory syndrome induced by adjuvants; CI, confidence interval; COVID-19, Coronavirus Disease 2019; DH, Department of Health; DKA, diabetic ketoacidosis; GOPC, general outpatient clinic; HA, Hong Kong Hospital Authority; HR, hazard ratio; ICD-9-CM, International Classification of Diseases, Ninth Revision, Clinical Modification; ICPC, International Classification of Primary Care; IPTW, inverse probability of treatment weighting; IQR, interquartile range; IRR, incidence rate ratio; NA, not applicable; NSAID, nonsteroidal anti-inflammatory drug; PH, proportional hazard; RAT, rapid antigen test; RMST, restricted mean survival time; RMTL, restricted mean time lost; RT-PCR, reverse transcription polymerase chain reaction; SARS-CoV-2, Severe Acute Respiratory Syndrome Coronavirus 2; SD, standard deviation; SMD, standardised mean difference; SOPC, specialist outpatient clinic; STROBE, Strengthening the Reporting of Observational Studies in Epidemiology.

## What did the researchers do and find?

- This study included 167,337 CoronaVac, 158,378 BNT162b2 recipients, and 145,199 COVID-19 patients with their respective 1:1 matched control.

- There was no evidence of increased risks of incident diabetes following COVID-19 vaccination.

- Regardless of predominant circulating variants, SARS-CoV-2 infection was associated with significantly higher risks of incident diabetes, particularly type 2 diabetes. However, these risks were lower with Omicron variants.

- Fully vaccinated COVID-19 survivors did not have an increased risk of incident diabetes.

## What do these findings mean?

- There is still an increased risk of incident diabetes following SARS-CoV-2 infection even with the prevailing Omicron variants, although the risk is lower.

- Fully vaccinated individuals might be protected from the risk of incident diabetes following SARS-CoV-2 infection.

- As there was no evidence of increased risks of incident diabetes following COVID-19 vaccination, our results encourage people to get fully vaccinated to protect themselves from severe complications of COVID-19 and the sequelae of long COVID, including the potential risk of incident diabetes.

- Causal interpretation of these findings is limited by potential misclassification bias as type 1 diabetes was identified through diagnostic coding and possible residual confounders.

## Introduction

The pandemic of Coronavirus Disease 2019 (COVID-19), caused by Severe Acute Respiratory Syndrome Coronavirus 2 (SARS-CoV-2), has infected more than 650 million people worldwide, causing more than 6.6 million deaths globally at the time of writing [1]. The expression of angiotensin-converting enzyme 2 (ACE2), the entry receptor for SARS-CoV-2, has been found not only in the respiratory system but also in extrapulmonary systems including the pancreas [2], raising concerns about new-onset diabetes after SARS-CoV-2 infection from the potential direct effect on pancreatic beta cells [3].

Several nationwide cohorts have investigated into this issue [4]. A cohort using the database from the United States (US) Department of Veteran Affairs showed 40% excess risk of incident diabetes with a median follow-up of 1 year, mostly type 2 diabetes [5]. Similarly, a study using German nationwide database showed 30% excess risk of incident type 2 diabetes after SARS-CoV-2 infection with a median follow-up duration of 4 months, compared to other acute respiratory illness [6]. However, a cohort study in the United Kingdom (UK) showed that the increased risk of incident diabetes persisted up to 12 weeks after SARS-CoV-2 infection [7]. Of

note, there are differences between Asians and Caucasians in the pathophysiology of type 2 diabetes. At any given BMI, compared to Caucasians, Asians have more visceral adiposity, which is metabolically more adverse and contributes to lipotoxicity and insulin resistance [8]. This interethnic difference can modify the risks of incident diabetes after SARS-CoV-2 infection, which remains to be elucidated among Asians. Furthermore, with the emergence of SARS-CoV-2 variants where Omicron variants are the predominant strains at the time of writing, it remains to be determined whether the risks of incident diabetes following SARS-CoV-2 might differ compared with previous variants. There were indeed suggestions of fewer long COVID symptoms and burdens with Omicron variants compared to previous variants [9,10].

COVID-19 vaccination has shown efficacy in reducing severe disease, developed based on different technology platforms. At the time of writing, more than 12 billion doses of COVID-19 vaccination have been administered [1]. At least 15 cases of type 1 diabetes have been reported after both mRNA and inactivated COVID-19 vaccines [11]. There were also cases of acute hyperglycaemic crises, either on a background of known type 2 diabetes or as the first presentation of type 2 diabetes, following COVID-19 vaccination [12]. Autoimmune/inflammatory syndrome induced by adjuvants (ASIA) and molecular mimicry are among the postulated mechanisms [11]. Little is known regarding the glycaemic status of nondiabetic individuals around the time of COVID-19 vaccination. Besides, case reports and series do not quantify the absolute risk of incident diabetes and inform if COVID-19 vaccination is associated with an increased risk of new-onset diabetes, especially since onset of type 2 diabetes after COVID-19 vaccination is expected to be underreported. Such information is essential to inform clinical practice and guide patients regarding COVID-19 vaccine uptake and the subsequent follow-up and monitoring.

It has been shown that COVID-19 vaccination may decrease the severity of symptoms of long COVID [13]. Following the same line of thought, COVID-19 vaccination may modify the risk trajectory of incident diabetes following SARS-CoV-2 infection. It is important to understand the role of COVID-19 vaccination in the risk of new-onset diabetes after SARS-CoV-2 infection, as the number of COVID-19 survivors keeps increasing globally.

As of November 2022, the Hong Kong Government Vaccination Programme provides 2 main authorised COVID-19 vaccines: CoronaVac (inactivated whole-virus vaccine) from Sinovac Biotech (Hong Kong) Limited and BNT162b2 (monovalent mRNA vaccine) from BioNTech/Fosun Pharma in China (equivalent to the Pfizer-BioNTech vaccine outside China) [14,15]. Hence, we conducted this population-based study to evaluate the incidence of diabetes (i) associated with COVID-19 vaccination, and (ii) following SARS-CoV-2 infection, with further stratifications according to vaccination status and SARS-CoV-2 variants.

## Methods

### Ethics statement

The study protocol was approved by the Institutional Review Board of the University of Hong Kong/Hospital Authority Hong Kong West Cluster (UW 20–556, UW 21–149, and UW 21–138); the Central Institutional Review Board of the Hospital Authority of Hong Kong (CIRB-2021-005-4); and the Department of Health Ethics Committee (LM 21/2021, LM171/2021, and LM 175/2022). Informed patient consent was not required as the data used in this study were anonymised.

### Data source

The anonymised, population-wide COVID-19 vaccination records were available from the Department of Health (DH) of Hong Kong, and electronic medical records were provided by

the Hong Kong Hospital Authority (HA). These 2 databases are linked based on hashed unique identifiers. Vaccination records included the date of administration and the types of vaccines. HA, a statutory administrative institution, provides public healthcare services to more than 7.3 million Hong Kong residents covering approximately 90% of all primary, secondary, and tertiary care services in Hong Kong [16]. All individual data entered into each EMR system across 43 public hospitals, 49 specialist outpatient clinics (SOPCs), and 73 general outpatient clinics (GOPCs) are gathered by HA [17]. These centralised medical records included demographics, date of registered death, drug dispensing records, diagnoses, procedures, and laboratory tests. The linked vaccine safety data have been comprehensively used to conduct pharmacovigilance studies of COVID-19 vaccines [18–28]. The study is reported as per the Strengthening the Reporting of Observational Studies in Epidemiology (STROBE) guideline (S1 Checklist). Furthermore, the data collection and analysis were based on a prospective protocol (S1 Protocol) [29].

## Study design and population

These 2 cohorts were identified separately to evaluate the risks of diabetes following COVID-19 vaccination and SARS-CoV-2 infection. We identified people (except for pregnant women) who ever used HA services since 2018.

## The first cohort for evaluating risks following COVID-19 vaccination (the first cohort)

The first cohort included people who received at least 1 dose of COVID-19 vaccine (separately performed for BNT162b2 and CoronaVac) from February 23, 2021, to September 30, 2021, and those who did not receive any COVID-19 vaccines up to September 30, 2021. The COVID-19 vaccination policy and vaccine uptake rate in Hong Kong have been elaborated in S1 Method. The inclusion period was chosen to minimise the potential modifying effect of SARS-CoV-2 infection before vaccination, given the low number of new COVID-19 cases during that time in Hong Kong. The date of the first dose was used as the index date for vaccination recipients. To assign pseudo-index dates for unvaccinated people, we matched them with vaccine recipients based on sex and age. To ensure each unvaccinated person could match with a vaccine recipient, the maximum ratio was used. Afterwards, the index date of vaccine recipients was assigned to corresponding unvaccinated people. [18,22,24,26]. They were followed up from the index date until a diagnosis of the outcome, death, date of SARS-CoV-2 infection or the end of the study period (August 15, 2022), whichever occurred first.

## The second cohort for evaluating risks following SARS-CoV-2 infection (the second cohort)

The second cohort consisted of COVID-19 patients identified by a first positive result on the SARS-CoV-2 reverse transcription polymerase chain reaction (RT-PCR) test or rapid antigen test (RAT) from January 1, 2020, to March 31, 2022, and people who were never infected up to March 31, 2022. The inclusion period was chosen to capture all COVID-19 cases from the date of the first confirmed cases to the peak of the Omicron wave in Hong Kong. The COVID-19 pandemic and key policies in Hong Kong have been described in S2 Method. The index date was defined as the first date of SARS-CoV-2 infection for COVID-19 patients. To assign pseudo-index dates for people without infection, we matched them with COVID-19 patients based on sex and age. To ensure each non-COVID-19 person could match with a COVID-19 patient, the maximum ratio was used. The index date of COVID-19 patient was then assigned

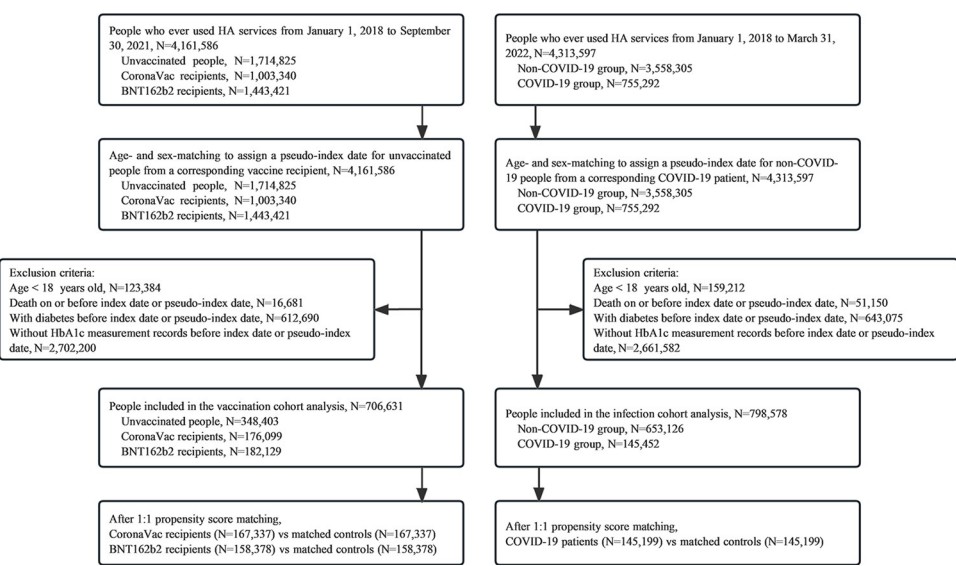

**Fig 1. Flowchart of record linkage and selection procedure.**

to corresponding non-COVID-19 people. This cohort was followed up till the occurrence of outcomes, death, or August 15, 2022, whichever came first.

Exclusion criteria were (i) age <18 years, (ii) individuals who died on or before the index or pseudo-index date, (iii) individuals who had diabetes before the index or pseudo-index date, and (iv) individuals without HbA1c measurement records before the index or pseudo-index date (Fig 1).

## Outcome definition

Outcomes of interest include overall diabetes, type 2 diabetes, and type 1 diabetes. The diagnosis of diabetes was adapted from a previous study [5]: the International Classification of Diseases, Ninth Revision, Clinical Modification (ICD-9-CM) diagnosis codes (250.XX), or the International Classification of Primary Care (ICPC) codes (T89 or T90), or an HbA1c measurement of ≥6.5% (48 mmol/mol), or a prescription record of diabetes medication for more than 30 days (British National Formulary codes: 6.1.1.X or 6.1.2.X). Type 1 diabetes was defined based on ICD-9-CM (250.x1 or 250.x3) and ICPC codes (T89). In view of case reports of acute hyperglycaemic crises following COVID-19 vaccination, we also retrieved the events of acute hyperglycaemia (ICD-9-CM: 250.82 to 250.83, 250.20, and 250.22 to 250.23) and diabetic ketoacidosis (DKA) (ICD-9-CM: 250.10, 250.12 to 250.13, and 250.30 to 250.33).

## Statistical analysis

We conducted a propensity score matched cohort study, matching participants and controls at a 1:1 ratio using the propensity score (see S3 Method for details). Baseline characteristics before and after propensity score matching were presented as means with standard deviation (SD) for continuous variables and frequencies with percentages for categorical variables. The association between incident diabetes and COVID-19 vaccination was estimated through Cox proportional hazards (PHs) regression. Analyses were repeated to estimate the association between incident diabetes and SARS-CoV-2 infection. Crude incidence rates per 100,000 person-days were reported for vaccination recipients, COVID-19 patients, and their respective control groups. The number needed to harm was calculated as the reciprocal of the difference

in cumulative incidence rates between 2 groups, indicating the number of participants who need to be exposed at a time point for 1 additional diabetes case. Then, we evaluated the hazard ratios (HRs) and corresponding 95% confidence intervals (CIs) of incident diabetes. To account of the pair-matched structure in the data, we used a cluster-robust sandwich variance–covariance estimator in all Cox regression models.

For both cohorts, separate Cox regression models were conducted to assess the risk of overall diabetes by subgroups of age (<60 and ≥60 years), sex, and prediabetes. To evaluate if the association between COVID-19 vaccination and incident diabetes was modified by previous SARS-CoV-2 infection, we performed subgroup analyses by history of SARS-CoV-2 infection. To evaluate if the association between SARS-CoV-2 infection and incident diabetes was modified by vaccination or predominant variants, we also performed subgroup analyses by vaccination status and stratified the participants into those who were infected before and those who were infected during the Omicron wave (since January 1, 2022). HRs with 95% CIs were calculated for each subgroup analysis. *P* values for interaction terms were calculated for each stratifying variable.

Furthermore, 2 sensitivity analyses were conducted. First, we compared the incidence of diabetes following 2 doses of either CoronaVac or BNT162b2 with their 1:1 matched unvaccinated control. Second, people were censored at the date of vaccination to remove potential modification effects by vaccination in the analysis of evaluating risks following SARS-CoV-2 infection.

We performed 4 post hoc sensitivity analyses. First, the Bonferroni correction was applied to account for multiple hypotheses testing. We set the α level (probability of type I error) to 0.05/15 (number of analyses) = 0.00333 and calculated Bonferroni-corrected CIs for HRs [30]. Second, we performed the restricted mean survival time (RMST) analysis, which is suggested as a supplement to the Cox PH model analysis without relying on the PH assumption [31,32]. Compared with the Cox PH model analysis, RMST offered more interpretable metrics, which have been recently applied in a variety of domains, including evaluating the treatment effect in people with type 2 diabetes [33,34]. The RSMT difference and restricted mean time lost (RMTL) ratio for each outcome were calculated. The RMST difference refers to the average event-free survival time difference over a restricted time horizon. RMTL represents the region above the Kaplan–Meier survival curve and indicates the average event-free survival time lost up to a restricted time horizon. The HR is proposed to be approximated using the RMTL ratio between treatment groups without imposing the PH assumption [32]. In our analysis, the time horizon for a given outcome was determined as the minimum of the largest followed-up time (99%) of vaccine recipients/COVID-19 patients and their matched controls. Third, we evaluated the associations between COVID-19 vaccination/infection and incident diabetes using Poisson regression after propensity score matching. The incidence rate ratios (IRRs) and corresponding 95% CIs of incident diabetes were estimated to compare the risks between COVID-19 vaccination/infection and their respective matched controls. Fourth, we used the propensity score to perform inverse probability of treatment weighting (IPTW) and truncated the weight at the first and 99th percentile of the observed PS weighting distribution to address extreme weights. Cox regression models weighted by the IPTW were fitted to estimate the risks of incident diabetes following COVID-19 vaccination and infection.

Following the recommendation of the American Statistical Association [35], we presented 95% of two-sided CIs and interpreted the results based on point estimates with their respective CIs. All statistical analyses were performed using the Stata Version 16.0 (StataCorp LP, College Station, TX). The analyses were conducted by XX and analysed independently by MC and ICHA for quality assurance.

## Results

### Baseline characteristics

In total, we identified 706,631 people in the first cohort (CoronaVac: *n* = 176,099; BNT162b2: *n* = 182,129; unvaccinated: *n* = 348,403) and 798,578 people in the second cohort (COVID-19 patients, *n* = 145,452; non-COVID-19 group, *n* = 653,126) (S3 Table). After excluding those whom we could not identify matched pairs, we included 167,337 CoronaVac recipients and 167,337 matched controls, 158,378 BNT162b2 recipients and 158,378 matched controls, in addition to 145,199 COVID-19 patients and 145,199 matched controls.

In the first cohort, compared with the controls, the vaccination recipients were younger and less likely to have comorbidities. Among the vaccination recipients, 87,754 (52.4%) CoronaVac recipients and 73,500 (46.4%) BNT162b2 recipients have prediabetes, respectively. In the second cohort, compared with controls, COVID-19 survivors were older and had more comorbidities. Among the COVID-19 survivors, 60,348 (41.6%) were fully vaccinated and 25,792 (17.8%) did not receive any COVID-19 vaccines. After matching, the propensity score distributions were highly overlapping (S1 Fig) and the baseline characteristics were balanced between the study population and matched controls, with all SMDs <0.1 (Table 1).

### Risk of incident diabetes following COVID-19 vaccination

In the first cohort, we identified 5,760 and 4,411 diabetes cases after receiving CoronaVac and BNT162b2 vaccines, respectively. CoronaVac recipients were followed up for a median of 386 days (interquartile range [IQR] 341 to 435) and BNT162b2 recipients for a median of 384 days (IQR 347 to 429). The median duration from vaccination to onset of diabetes was 178 days (IQR 91 to 283) and 179 days (IQR 91 to 288) for CoronaVac and BNT162b2 recipients, respectively. The cumulative incidences of overall diabetes, type 2 diabetes, and type 1 diabetes between the vaccine recipients and respective control groups are shown in S2 Fig. The crude incidence rates of overall diabetes were 9.08 per 100,000 person-days among CoronaVac recipients and 9.10 per 100,000 person-days among matched controls, and 7.41 per 100,000 person-days among BNT162b2 recipients and 8.58 per 100,000 person-days among matched controls. Compared to matched unvaccinated people, there was no evidence of the associations of CoronaVac or BNT162b2 vaccination with increased risks of overall diabetes (CoronaVac: HR = 0.998 [95% CI 0.962 to 1.035]; BNT162b2: HR = 0.862 [95%CI 0.828 to 0.897]), type 2 diabetes (CoronaVac: HR = 0.997 [95% CI 0.962 to 1.035]; BNT162b2: HR = 0.862 [95% CI 0.828 to 0.898]), and type 1 diabetes (CoronaVac: HR = 1.337 [95% CI 0.300 to 5.949]; BNT162b2: HR = 0.511 [95% CI 0.092 to 2.846]) (Table 2). Among CoronaVac and BNT162b2 recipients, 17 and 9 cases of acute hyperglycaemia and 2 and 5 cases of DKA were observed, respectively. We repeated the subgroup analyses according to age, sex, prediabetes, and previous SARS-CoV-2 infection, which showed consistent results for both types of COVID-19 vaccines (Fig 2). The results of sensitivity analyses that compared the incidence of diabetes following 2 doses of either vaccine with unvaccinated control were consistent with the result of the main analysis (S4 Table).

### Risk of incident diabetes following SARS-CoV-2 infection

In the second cohort, we identified 2,109 cases of diabetes following SARS-CoV-2 infection. COVID-19 patients were followed up for a median of 164 days (IQR 155 to 168). The median duration from SARS-CoV-2 infection to onset of diabetes was 55 days (IQR 18 to 118). The cumulative incidences of overall diabetes and type 2 diabetes among COVID-19 patients and controls are shown in S4 Fig. There were 2,109 (1.45%) patients diagnosed with diabetes

**Table 1. Baseline characteristics of BNT162b2 or CoronaVac recipients, COVID-19 patients, and their respective matched controls after 1:1 propensity score matching.**

| Baseline characteristics | CoronaVac (N = 167,337) | Control (N = 167,337) | SMD | BNT162b2 (N = 158,378) | Control (N = 158,378) | SMD | COVID-19 patients (N = 145,199) | Control (N = 145,199) | SMD |
|---|---|---|---|---|---|---|---|---|---|
| | Mean ± SD /N (%) | Mean ± SD /N (%) | | Mean ± SD /N (%) | Mean ± SD /N (%) | | Mean ± SD /N (%) | Mean ± SD /N (%) | |
| Age, years | 61.9 ± 12.2 | 63.9 ± 14.5 | | 57.3 ± 14.0 | 59.4 ± 15.7 | | 62.7 ± 16.0 | 61.9 ± 15.2 | |
| 18–44 | 14,411 (8.6) | 14,563 (8.7) | 0.03 | 29,465 (18.6) | 28,589 (18.1) | 0.04 | 20,094 (13.8) | 19,349 (13.3) | 0.04 |
| 45–59 | 50,712 (30.3) | 47,240 (28.2) | | 50,401 (31.8) | 47,505 (30.0) | | 37,267 (25.7) | 39,544 (27.2) | |
| ≥60 | 102,214 (61.1) | 105,534 (63.1) | | 78,512 (49.6) | 82,284 (52.0) | | 87,838 (60.5) | 86,306 (59.4) | |
| Sex | | | | | | | | | |
| Male | 77,980 (46.6) | 76,931 (46.0) | 0.01 | 70,743 (44.7) | 70,011 (44.2) | 0.01 | 66,956 (46.1) | 65,454 (45.1) | 0.02 |
| Female | 89,357 (53.4) | 90,406 (54.0) | | 87,635 (55.3) | 88,367 (55.8) | | 78,243 (53.9) | 79,745 (54.9) | |
| Previous SARS-CoV-2 infection | 442 (0.3) | 424 (0.3) | 0.00 | 693 (0.4) | 639 (0.4) | 0.01 | NA | NA | NA |
| Vaccination status | | | | | | | | | |
| Unvaccinated | 0 (0.0) | 167,337 (100.0) | NA | 0 (0.0) | 158,378 (100.0) | NA | 25,792 (17.8) | 26,111 (18.0) | 0.04 |
| Partially vaccinated* | 167,337 (100.0) | 0 (0.0) | NA | 158,378 (100.0) | 0 (0.0) | NA | 59,059 (40.7) | 54,268 (37.4) | |
| Fully vaccinated† | 0 (0.0) | 0 (0.0) | NA | 0 (0.0) | 0 (0.0) | NA | 60,348 (41.6) | 64,820 (44.6) | |
| Prediabetes‡ | 87,754 (52.4) | 87,163 (52.1) | 0.01 | 73,500 (46.4) | 73,958 (46.7) | 0.01 | 70,500 (48.6) | 72,197 (49.7) | 0.02 |
| Preexisting comorbidities | | | | | | | | | |
| Charlson Comorbidity Index | 2.8 ± 1.3 | 2.9 ± 1.4 | | 2.4 ± 1.4 | 2.5 ± 1.5 | | 3.0 ± 1.8 | 2.9 ± 1.7 | |
| 0 | 8,235 (4.9) | 8,327 (5.0) | 0.02 | 20,888 (13.2) | 20,085 (12.7) | 0.02 | 13,468 (9.3) | 13,108 (9.0) | 0.01 |
| 1–2 | 56,701 (33.9) | 54,574 (32.6) | | 58,602 (37.0) | 57,526 (36.3) | | 43,251 (29.8) | 44,521 (30.7) | |
| ≥3 | 102,401 (61.2) | 104,436 (62.4) | | 78,888 (49.8) | 80,767 (51.0) | | 88,480 (60.9) | 87,570 (60.3) | |
| Myocardial infarction | 1,471 (0.9) | 1,576 (0.9) | 0.01 | 1,094 (0.7) | 1,143 (0.7) | 0.00 | 2,303 (1.6) | 2,183 (1.5) | 0.01 |
| Peripheral vascular disease | 411 (0.2) | 423 (0.3) | 0.00 | 305 (0.2) | 325 (0.2) | 0.00 | 785 (0.5) | 748 (0.5) | 0.00 |
| Cerebrovascular disease | 8,798 (5.3) | 9,405 (5.6) | 0.02 | 5,938 (3.7) | 6,192 (3.9) | 0.01 | 12,431 (8.6) | 11,804 (8.1) | 0.02 |
| Chronic obstructive pulmonary disease | 3,888 (2.3) | 4,111 (2.5) | 0.01 | 3,317 (2.1) | 3,320 (2.1) | 0.00 | 5,689 (3.9) | 5,185 (3.6) | 0.02 |
| Dementia | 367 (0.2) | 388 (0.2) | 0.00 | 126 (0.1) | 127 (0.1) | 0.00 | 1,694 (1.2) | 1,364 (0.9) | 0.02 |
| Paralysis | 370 (0.2) | 411 (0.2) | 0.01 | 212 (0.1) | 249 (0.2) | 0.01 | 823 (0.6) | 732 (0.5) | 0.01 |
| Chronic renal failure | 1,832 (1.1) | 1,942 (1.2) | 0.01 | 1,385 (0.9) | 1,478 (0.9) | 0.01 | 3,302 (2.3) | 2,839 (2.0) | 0.02 |
| Mild liver disease | 302 (0.2) | 325 (0.2) | 0.00 | 266 (0.2) | 277 (0.2) | 0.00 | 451 (0.3) | 434 (0.3) | 0.00 |
| Moderate–severe liver disease | 167 (0.1) | 182 (0.1) | 0.00 | 146 (0.1) | 147 (0.1) | 0.00 | 310 (0.2) | 259 (0.2) | 0.01 |
| Ulcers | 1,809 (1.1) | 1,839 (1.1) | 0.00 | 1,303 (0.8) | 1,252 (0.8) | 0.00 | 2,205 (1.5) | 2,069 (1.4) | 0.01 |
| Rheumatoid arthritis and other inflammatory polyarthropathies | 1,279 (0.8) | 1,383 (0.8) | 0.01 | 1,506 (1.0) | 1,567 (1.0) | 0.00 | 1,614 (1.1) | 1,631 (1.1) | 0.00 |
| Malignancy | 3,435 (2.1) | 3,615 (2.2) | 0.01 | 3,305 (2.1) | 3,523 (2.2) | 0.01 | 5,503 (3.8) | 5,294 (3.6) | 0.01 |
| Metastatic solid tumour | 355 (0.2) | 387 (0.2) | 0.00 | 304 (0.2) | 315 (0.2) | 0.00 | 1,063 (0.7) | 990 (0.7) | 0.01 |
| Mental disorders | 11,102 (6.6) | 11,629 (6.9) | 0.01 | 10,395 (6.6) | 10,667 (6.7) | 0.01 | 15,273 (10.5) | 13,939 (9.6) | 0.03 |
| Obesity | 9,411 (5.6) | 9,414 (5.6) | 0.00 | 8,044 (5.1) | 8,073 (5.1) | 0.00 | 8,461 (5.8) | 7,809 (5.4) | 0.02 |
| Use of medications within 90 days before index date | | | | | | | | | |
| Renin-angiotensin-system agents | 35,822 (21.4) | 36,543 (21.8) | 0.01 | 29,846 (18.8) | 30,486 (19.2) | 0.01 | 25,183 (17.3) | 24,353 (16.8) | 0.02 |
| Beta blockers | 27,919 (16.7) | 29,134 (17.4) | 0.02 | 22,829 (14.4) | 23,955 (15.1) | 0.02 | 19,668 (13.5) | 18,995 (13.1) | 0.01 |
| Calcium channel blockers | 61,002 (36.5) | 61,316 (36.6) | 0.00 | 48,236 (30.5) | 48,660 (30.7) | 0.01 | 41,497 (28.6) | 40,251 (27.7) | 0.02 |
| Diuretics | 6,689 (4.0) | 7,006 (4.2) | 0.01 | 5,320 (3.4) | 5,450 (3.4) | 0.00 | 7,862 (5.4) | 6,961 (4.8) | 0.03 |
| Nitrates | 7,716 (4.6) | 8,032 (4.8) | 0.01 | 5,897 (3.7) | 6,241 (3.9) | 0.01 | 5,821 (4.0) | 5,385 (3.7) | 0.02 |

*(Continued)*

**Table 1.** (Continued)

| Baseline characteristics | CoronaVac (N = 167,337) | Control (N = 167,337) | SMD | BNT162b2 (N = 158,378) | Control (N = 158,378) | SMD | COVID-19 patients (N = 145,199) | Control (N = 145,199) | SMD |
|---|---|---|---|---|---|---|---|---|---|
| | Mean ± SD /N (%) | Mean ± SD /N (%) | | Mean ± SD /N (%) | Mean ± SD /N (%) | | Mean ± SD /N (%) | Mean ± SD /N (%) | |
| Lipid-lowering agents | 56,440 (33.7) | 57,486 (34.4) | 0.01 | 46,310 (29.2) | 47,445 (30.0) | 0.02 | 36,592 (25.2) | 37,126 (25.6) | 0.01 |
| Antiarrhythmic drugs | 600 (0.4) | 614 (0.4) | 0.00 | 560 (0.4) | 581 (0.4) | 0.00 | 639 (0.4) | 606 (0.4) | 0.00 |
| Cardiac glycosides | 785 (0.5) | 862 (0.5) | 0.01 | 544 (0.3) | 556 (0.4) | 0.00 | 1,164 (0.8) | 1,053 (0.7) | 0.01 |
| Anticoagulants | 3,182 (1.9) | 3,337 (2.0) | 0.01 | 2,341 (1.5) | 2,461 (1.6) | 0.01 | 4,492 (3.1) | 4,160 (2.9) | 0.01 |
| Antiplatelets | 28,732 (17.2) | 29,785 (17.8) | 0.02 | 21,668 (13.7) | 22,634 (14.3) | 0.02 | 21,024 (14.5) | 20,040 (13.8) | 0.02 |
| Antifibrinolytics and haemostatics | 1,040 (0.6) | 995 (0.6) | 0.00 | 1,220 (0.8) | 1,197 (0.8) | 0.00 | 1,054 (0.7) | 966 (0.7) | 0.01 |
| Hormonal therapy | 2,030 (1.2) | 2,191 (1.3) | 0.01 | 2,491 (1.6) | 2,507 (1.6) | 0.00 | 1,637 (1.1) | 1,532 (1.1) | 0.01 |
| Glucocorticoids | 2,655 (1.6) | 2,874 (1.7) | 0.01 | 3,346 (2.1) | 3,470 (2.2) | 0.01 | 3,998 (2.8) | 3,669 (2.5) | 0.01 |
| Antidepressants | 10,566 (6.3) | 11,126 (6.6) | 0.01 | 10,470 (6.6) | 10,898 (6.9) | 0.01 | 8,151 (5.6) | 7,880 (5.4) | 0.01 |
| NSAIDs | 15,314 (9.2) | 14,907 (8.9) | 0.01 | 15,202 (9.6) | 14,903 (9.4) | 0.01 | 9,575 (6.6) | 8,946 (6.2) | 0.02 |
| Drugs for gout | 5,988 (3.6) | 6,052 (3.6) | 0.00 | 4,912 (3.1) | 5,025 (3.2) | 0.00 | 4,827 (3.3) | 4,385 (3.0) | 0.02 |
| Antiepileptic drugs | 5,339 (3.2) | 5,708 (3.4) | 0.01 | 5,878 (3.7) | 6,259 (4.0) | 0.01 | 5,620 (3.9) | 5,235 (3.6) | 0.01 |
| Antiviral drugs | 4,961 (3.0) | 4,888 (2.9) | 0.00 | 4,660 (2.9) | 4,593 (2.9) | 0.00 | 3,996 (2.8) | 3,403 (2.3) | 0.03 |
| Antibacterial drugs | 8,173 (4.9) | 8,280 (4.9) | 0.00 | 8,285 (5.2) | 8,182 (5.2) | 0.00 | 12,177 (8.4) | 10,381 (7.1) | 0.05 |
| Immunosuppressants | 1,448 (0.9) | 1,569 (0.9) | 0.01 | 2,147 (1.4) | 2,256 (1.4) | 0.01 | 1,629 (1.1) | 1,623 (1.1) | 0.00 |

COVID-19, Coronavirus Disease 2019; NA, not applicable; NSAID, nonsteroidal anti-inflammatory drug; SD, standard deviation; SMD, standardised mean difference.

*Partially vaccinated people were defined as vaccine recipients who received 1 dose of BNT162b2 or no more than 2 doses of CoronaVac.

†Fully vaccinated people were defined as those with at least 2 doses of BNT162b2 or 3 doses of CoronaVac.

‡Prediabetes was defined as baseline HbA1c ≥5.7% and <6.5%.

following SARS-CoV-2 infection. No event of type 1 diabetes was identified among adult COVID-19 survivors. SARS-CoV-2 infection was associated with a significantly higher risk of overall diabetes (9.04 versus 7.38 per 100,000 person-days, HR = 1.225 [95% CI 1.150 to 1.305]) and type 2 diabetes (9.04 versus 7.38 per 100,000 person-days, HR = 1.226 [95% CI 1.151 to 1.306]) compared with matched controls (Table 2). The number needed to harm at 6 months was 406 for 1 additional diabetes case and 404 for 1 additional type 2 diabetes case. There were 16 and 4 cases of acute hyperglycaemia and DKA following SARS-CoV-2 infection. Results of subgroup analyses by sex and prediabetes were consistent with the primary analysis. In the subgroup analyses stratified by age and vaccination status, we found no evidence of the associations between SARS-CoV-2 infection and an increased risk of diabetes among patients aged <60 years (HR = 1.070 [95% CI 0.952 to 1.202]) and who were fully vaccinated (HR = 1.005 [95% CI 0.904 to 1.116]). The associations between increased risks of incident diabetes and SARS-CoV-2 infection were stronger in older (HR = 1.292 [95% CI 1.199 to 1.393], p for interaction < 0.001), unvaccinated people (HR = 1.694 [95% CI 1.484 to 1.933], p for interaction < 0.001) and people without prediabetes (HR = 1.598 [95% CI 1.326 to 1.926], p for interaction = 0.005). SARS-CoV-2 infection was associated with an increased risk of diabetes regardless of the predominant circulating variant (non-Omicron: HR = 1.871 [95% CI 1.352 to 2.589]; Omicron: HR = 1.209 [95% CI 1.134 to 1.289]) (Fig 2). Nonetheless, the excess risk was lower with Omicron variant compared with previous variants (p for interaction = 0.009). Results were consistent with the main analysis in sensitivity analyses after censoring at the date of vaccination (S5 Table).

**Table 2. Crude incidence rate of outcomes for CoronaVac or BNT162b2 recipients, COVID-19 patients, and respective matched controls, and HR of events for CoronaVac or BNT162b2 recipients and COVID-19 patients in comparison with their respective matched controls.**

| Events | Vaccine recipients or COVID-19 patients | | | | Control | | | | HR[†] | 95% CI | P value |
|---|---|---|---|---|---|---|---|---|---|---|---|
| | Cases with event | Crude incidence rate* | 95% CI | Person-days | Cases with event | Crude incidence rate* | 95% CI | Person-days | | | |
| **CoronaVac recipients vs. controls** | | | | | | | | | | | |
| Overall diabetes | 5,760 | 9.08 | (8.85, 9.32) | 63,430,730 | 5,771 | 9.10 | (8.87, 9.34) | 63,384,197 | 0.998 | (0.962, 1.035) | 0.899 |
| Type 2 diabetes | 5,756 | 9.07 | (8.84, 9.31) | 63,431,741 | 5,768 | 9.10 | (8.87, 9.34) | 63,384,855 | 0.997 | (0.962, 1.035) | 0.892 |
| Type 1 diabetes | 4 | 0.006 | (0.00, 0.02) | 64,660,279 | 3 | 0.00 | (0.00, 0.01) | 64,636,412 | 1.337 | (0.300, 5.949) | 0.703 |
| **BNT162b2 recipients vs. controls** | | | | | | | | | | | |
| Overall diabetes | 4,411 | 7.41 | (7.19, 7.63) | 59,532,322 | 5,154 | 8.58 | (8.35, 8.82) | 60,057,014 | 0.862 | (0.828, 0.897) | <0.001 |
| Type 2 diabetes | 4,409 | 7.41 | (7.19, 7.63) | 59,532,991 | 5,150 | 8.58 | (8.34, 8.81) | 60,057,786 | 0.862 | (0.828, 0.898) | <0.001 |
| Type 1 diabetes | 2 | 0.003 | (0.00, 0.01) | 60,435,717 | 4 | 0.01 | (0.00, 0.02) | 61,173,591 | 0.511 | (0.092, 2.846) | 0.443 |
| **COVID-19 patients vs. controls** | | | | | | | | | | | |
| Overall diabetes | 2,109 | 9.04 | (8.66, 9.44) | 23,324,634 | 1,775 | 7.38 | (7.04, 7.73) | 24,050,683 | 1.225 | (1.150, 1.305) | <0.001 |
| Type 2 diabetes | 2,109 | 9.04 | (8.66, 9.44) | 23,324,634 | 1,774 | 7.38 | (7.04, 7.73) | 24,050,697 | 1.226 | (1.151, 1.306) | <0.001 |
| Type 1 diabetes | 0 | 0.00 | NA | 23,544,532 | 1 | 0.00 | (0.00, 0.02) | 24,208,039 | NA | NA | NA |

CI, confidence interval; COVID-19, Coronavirus Disease 2019; HR, hazard ratio; NA, not applicable.

*The unit of crude incidence rate: events per 100,000 person-days.

[†]HR >1 (or <1) indicates vaccine recipients or COVID-19 patients had a higher risk (or lower risk) of outcome compared with their respective matched controls.

In the post hoc analysis, the results with Bonferroni correction were consistent with those without Bonferroni correction (S3 Fig). The results from the RMST analysis are shown in S6 Table. The RMST difference was consistent with the direction of the HRs from Cox regressions, and numerical values were close between HRs and RMTL ratios. The results of the analysis, which was performed using Poisson regression and IPTW-weighted Cox regression, were in line with the findings of the main analysis (S7 and S8 Tables).

## Discussion

Our study quantified the risks of incident diabetes after COVID-19 vaccination and evaluated the impact of COVID-19 vaccination and SARS-CoV-2 variants on the risks of incident diabetes after SARS-CoV-2 infection. There was no evidence of increased risks of incident diabetes following COVID-19 vaccination. On the other hand, we observed an increased risk of incident diabetes following SARS-CoV-2 infection. The increased risk was observed for type 2 diabetes, but not for type 1 diabetes. Interestingly, the observed excess risk of incident diabetes following SARS-CoV-2 infection was lower for Omicron variants compared with earlier variants. Furthermore, fully vaccinated individuals might be protected from the increased risk of diabetes following SARS-CoV-2 infection. Taken together, our results provide important

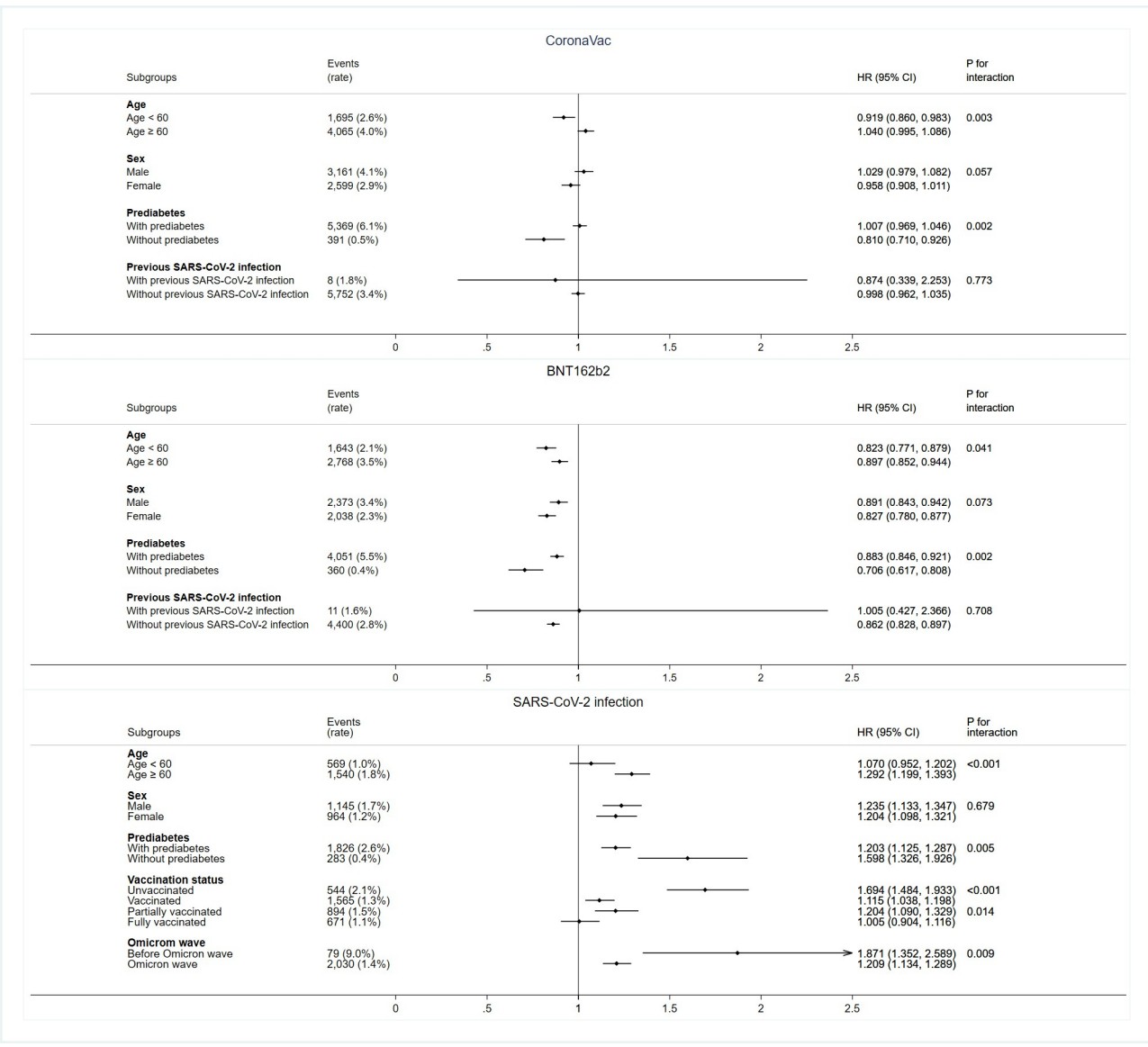

**Fig 2. Forest plot of HRs (95% CIs) for incident diabetes in different subgroup analyses.**

reassurance for adults to get fully vaccinated to protect themselves against adverse outcomes of SARS-CoV-2 infection and its long-term sequelae, including the risk of diabetes.

To date, incident diabetes following COVID-19 vaccination has only been reported in case reports and series, summarised by Pezzaioli and colleagues recently [11]. These were all cases of type 1 diabetes, aged between 27 and 73 years, mostly following mRNA vaccination, occasionally reported following inactivated and adenovirus-vectored vaccination. The time of onset varied from 3 days to 2 months postvaccination. In addition, a few further cases of acute hyperglycaemic crises have been described following COVID-19 vaccination, with relatively preserved C-peptide levels on presentation, raising suspicion of an entity of vaccine-induced hyperglycaemia [12,36]. Our cohort study clarified these concerns using a population-based dataset and appropriate unvaccinated controls, demonstrating no evidence of increased risks of incident diabetes following COVID-19 vaccination. For BNT162b2, the incidence rate of

diabetes was lower for vaccinated than for unvaccinated individuals. As our study was observational in nature, we could not firmly establish the causal relationship for BNT162b2 lowering the risk of incident diabetes. Unmeasured confounders could have accounted for this apparent protective effect, e.g., BNT162b2 recipients were younger and might have also adopted healthier lifestyle behaviours associated with the reduction in risks of incident diabetes. It is also possible that individuals with symptoms of undiagnosed diabetes were less likely to get vaccinated due to concerns about vaccine safety or efficacy.

In particular, the incidence of type 1 diabetes following COVID-19 vaccination was not increased and was in the order of 1 in 100,000, comparable to the background incidence rate of type 1 diabetes in Asian populations reported in the literature, which, in turn, is lower than that in the Caucasian populations [37]. Similarly, there was no increase in the risks of new-onset type 2 diabetes following COVID-19 vaccination. No studies have previously investigated the glycaemic status of nondiabetic individuals around the time of COVID-19 vaccination. On the other hand, there were studies of patients with diabetes for their perivaccination glycaemic control. Aberer and colleagues reported in patients with type 1 and type 2 diabetes monitored using continuous glucose monitoring system that time in range, below range, and above range did not substantially change following COVID-19 vaccination in the short term [38], consistent with the observations in our cohort of individuals without preexisting diabetes.

In contrast to the scant literature on risks of diabetes after COVID-19 vaccination, several large-scale cohort studies have assessed the risks of diabetes following SARS-CoV-2 infection, notably from the US [5], UK [7], and Germany [6]. The primary care database in Germany reported a 30% excess in risk of incident type 2 diabetes among COVID-19 survivors over a median follow-up of 4 months, but no significant increase in risk for other forms of diabetes [6]. A large cohort study in the US of over 180,000 COVID-19 survivors with a median follow-up of 1 year confirmed this increased risk of type 2 diabetes and further suggested that increasing COVID-19 severity was associated with higher risk of incident diabetes. Interestingly, this study reported that participants of African descent had a greater burden of incident diabetes than Caucasian participants, suggesting some interethnic differences in the propensity to develop incident diabetes after SARS-CoV-2 infection [5]. A more recent study utilising the UK healthcare database further divided the risk period into acute COVID-19 (4 weeks), postacute COVID (5 to 12 weeks), and long COVID (13 to 52 weeks) and showed that the risks of incident diabetes increase at least up to 12 weeks postacute COVID before declining. Consistent with the above cohorts, our study also showed that the risk of diabetes increased after SARS-CoV-2 infection. The increase in risk was seen in type 2 diabetes, but not for type 1 diabetes. Of note, the incidence of type 1 diabetes in Asians is lower than that in Caucasians [37]. Hence, the number of events of incident type 1 diabetes was low in our cohort, limiting the power of this analysis. Nonetheless, a post hoc analysis of 55 COVID-19 survivors evaluated at least 3 months after acute COVID-19 had detectable C-peptide levels, suggesting that insulinopenia was not apparent in postacute COVID-19 [39].

In all the aforementioned studies of risk of diabetes after SARS-CoV-2 infection, the inclusion criteria covered the period when COVID-19 vaccination programme commenced globally. Prior to our study, there was limited information regarding the influence of COVID-19 vaccination on the risk of incident diabetes following SARS-CoV-2 infection. Al-Aly and colleagues [40] showed that vaccinated COVID-19 survivors had a lower risk of long COVID compared with unvaccinated COVID-19 survivors, including metabolic disorders (encompassing "diabetes," "hyperlipidaemia," and "insulin use" in the study; HR = 0.77 [95% CI 0.68 to 0.87]), suggesting the benefits of COVID-19 vaccination on the risk of incident diabetes following SARS-CoV-2 infection. However, the HR for metabolic disorders following SARS-CoV-2 infection among vaccinated COVID-19 survivors was 1.32 (95% CI 1.26 to 1.39)

compared with vaccinated control groups [40]. Similarly, Kwan and colleagues reported lower diabetes risks after COVID-19 infection in vaccinated than in unvaccinated patients in a study done in the US, suggesting a benefit of vaccination [41]. The subgroup analysis in our current study stratified by vaccination status revealed a graded attenuation in risk of incident diabetes among COVID-19 survivors compared with their respective non-COVID controls upon completion of COVID-19 vaccination regime (unvaccinated: HR = 1.69 [95% CI 1.48 to 1.93], partially vaccinated: HR = 1.20 [95% CI 1.09 to 1.33], fully vaccinated: HR = 1.01 [95% CI 0.90 to 1.12]). Our results suggested that there was no significant increase in risk of incident diabetes following SARS-CoV-2 infection only among fully vaccinated individuals, but not for partially vaccinated or unvaccinated individuals. In fact, this may also explain the finding in the subgroup analysis that younger individuals did not have a significant increased risk of incident diabetes, since the proportion of fully vaccinated individuals were higher among younger individuals (55.9%) compared with older one (34.5%). The difference between our results and those reported by Al-Aly and colleagues [40] could be related to our study dedicated to the evaluation of glycaemic status, having matched for prediabetes status at baseline and requiring availability of HbA1c values in each individual prior to the entry of the cohort. Our results concurred with most studies that reported fewer symptoms postacute COVID-19 among the fully vaccinated individuals [42], adding to the literature regarding the benefits of COVID-19 vaccination in reducing incident diabetes as one of the potential manifestations of long COVID. Our results further suggested the consistent ability of various types of COVID-19 vaccination in reducing long COVID, as reviewed by Notarte and colleagues [42]. Possible mechanisms of protection of COVID-19 vaccination against incident diabetes included (i) reducing the severity of SARS-CoV-2 infection and (ii) hastening the clearance of SARS-CoV-2, which, in turn, reducing the exaggerated inflammatory responses in COVID-19.

In the subgroup analysis, we also noted an apparently stronger association between increased risks of incident diabetes and SARS-CoV-2 infection among individuals without prediabetes. This could be related to the inclusion criteria requiring an HbA1c measurement before an index date in all individuals. There could be several postulations for this observation: (i) individuals with prediabetes are already predisposed to the development of diabetes such that the additional impact of SARS-CoV-2 infection may be less significant; and (ii) individuals with prediabetes could have more likely taken lifestyle modifications to reduce their risk of developing diabetes.

With the evolution of the COVID-19 pandemic, Omicron variant of SARS-CoV-2 has become the dominant strain globally. In the stratified analysis, we observed that both non-Omicron and Omicron variants of SARS-CoV-2 were associated with increased risks of incident diabetes following SARS-CoV-2 infection. Interestingly, the excess risk of incident diabetes was lower for Omicron variants compared with non-Omicron variants. This was indeed in keeping with the observations from other studies reporting fewer symptoms of long COVID and less burden of long COVID with Omicron variants [9,10], which could be related to the less severe acute disease in the infection with SARS-CoV-2 Omicron variants [10].

The main strength of our study is that we have quantified the risk of new-onset diabetes following COVID-19 vaccination and further evaluated the risk of incident diabetes following SARS-CoV-2 infection considering the vaccination status and the prevalent SARS-CoV-2 variants using a population-based dataset. We were also able to analyse the risk of new-onset diabetes among recipients of 2 different types of COVID-19 vaccines. Nonetheless, our results should be interpreted bearing certain limitations. First, our results are not generalisable to recipients of types of COVID-19 vaccination other than inactivated and mRNA vaccination. BNT162b2 is currently in use in many jurisdictions around the world, including North America (e.g., US, Canada, Mexico), South America (e.g., Brazil, Chile), Europe (e.g., UK, Switzerland), and Asia (e.g., Japan, Singapore). The jurisdictions where CoronaVac is currently in use

are mostly located in Asia and Latin America, including China (where it was developed), Brazil, and Turkey. Second, information on BMI was not available from this cohort. Nonetheless, we have performed PS matching by including diagnosis of obesity in the analysis, and the balance was achieved with SMD <0.1. Third, the proportion of prediabetes in the current study cohort was around 50%, where the prevalence of prediabetes has been reported to be up to 40% among Chinese [43]. The slightly higher proportion of prediabetes individuals in the current study might be due to the requirement of valid HbA1c values before cohort entry for exclusion of individuals with preexisting diabetes, where HbA1c might have been checked among individuals with relatively higher cardiometabolic risks. Fourth, the modifying effects of differences in healthcare service utilisation on the risk of incident diabetes could not be entirely excluded. Individuals with SARS-CoV-2 infection may have more contact with healthcare services, resulting in more diagnostic tests and increased diagnosis of incident diabetes. On the other hand, since infection with Omicron variant is generally associated with less severe symptoms [44], while interaction with health services will be increased in those with symptoms, the effect will be less than with other SARS-CoV-2 variants. Likewise, vaccination reduces the severity of COVID-19 infections, so while those who were vaccinated still have an increased chance of being diagnosed to have diabetes after COVID-19 infection, the increase is less than in unvaccinated people. Fifth, type 1 diabetes was identified through diagnostic coding in the current study, where misclassification bias could not be entirely mitigated. Last but not least, the follow-up period was relatively short in our study to allow evaluation of the risk of chronic diabetic complications. The long-term impact of SARS-CoV-2 infection on diabetes requires continuous surveillance with global concerted efforts.

In conclusion, there was no evidence of increased risks of incident diabetes following COVID-19 vaccination. In contrast, the risk of incident diabetes increased following SARS-CoV-2 infection, especially for type 2 diabetes. This excess risk might be lower among survivors of SARS-CoV-2 infection with Omicron variants compared with previous variants. Fully vaccinated individuals might be protected from the risk of incident diabetes following SARS-CoV-2 infection. Our results should encourage people to get fully vaccinated to protect themselves from severe complications of COVID-19 and the sequelae of long COVID, including the potential risk of incident diabetes.

## Supporting information

**S1 Checklist. Strengthening the Reporting of Observational Studies in Epidemiology (STROBE) guideline.**
(DOC)

**S1 Table. ICD-9 Clinical Modification (CM) Codes used for disease identification.**
(DOCX)

**S2 Table. BNF codes used for medication.**
(DOCX)

**S3 Table. Baseline characteristics of BNT162b2 or CoronaVac recipients, unvaccinated people, COVID-19 patients, and non-COVID-19 people before propensity score matching.**
(DOCX)

**S4 Table. Crude incidence rate of outcomes for 2 doses of CoronaVac or BNT162b2 recipients and respective matched controls, and hazard ratio for 2 doses of CoronaVac or BNT162b2 recipients in comparison with their respective matched controls.**
(DOCX)

**S5 Table. Crude incidence rate of outcomes for COVID-19 patients and matched controls, and hazard ratio for COVID-19 patients in comparison with matched controls, censoring at the date of vaccination.**
(DOCX)

**S6 Table. The restricted mean survival time (RMST) difference and the restricted mean time lost (RMTL) ratio for outcomes.**
(DOCX)

**S7 Table. Crude incidence rate of outcomes for CoronaVac or BNT162b2 recipients, COVID-19 patients, and respective matched controls, and incidence rate ratio of events for CoronaVac or BNT162b2 recipients and COVID-19 patients in comparison with their respective matched controls.**
(DOCX)

**S8 Table. Crude incidence rate of outcomes for CoronaVac or BNT162b2 recipients, unvaccinated people, COVID-19 patients, and non-COVID-19 people before weighting, and hazard ratio after weighting.**
(DOCX)

**S1 Fig. Propensity score distributions for (a) CoronaVac recipients, (b) BNT162b2 recipients, and (c) COVID-19 patients and their respective matched controls before and after 1:1 propensity score matching.**
(TIF)

**S2 Fig. Cumulative incidence plots of various diabetes outcomes for CoronaVac recipient versus unvaccinated people, and BNT162b2 recipients versus unvaccinated people.**
(TIF)

**S3 Fig. Forest plots of hazard ratios with Bonferroni-corrected 95% CIs for incident diabetes for different hypothesis testing.**
(TIF)

**S4 Fig. Cumulative incidence plots of overall diabetes and type 2 diabetes outcomes for COVID-19 patients versus non-COVID-19 people.**
(TIF)

**S1 Method. COVID-19 vaccination policy in Hong Kong.**
(DOCX)

**S2 Method. COVID-19 pandemic and key policies in Hong Kong.**
(DOCX)

**S3 Method. Propensity score matching.**
(DOCX)

**S1 Protocol. Study protocol.**
(DOCX)

## Acknowledgments

The authors thank the Hospital Authority and the Department of Health for the generous provision of data for this study.

## Author Contributions

**Conceptualization:** Xi Xiong, David Tak Wai Lui, Carlos King Ho Wong.

**Data curation:** Xi Xiong, Carlos King Ho Wong.

**Formal analysis:** Xi Xiong, Carlos King Ho Wong.

**Funding acquisition:** Francisco Tsz Tsun Lai, Esther Wai Yin Chan, Ian Chi Kei Wong.

**Investigation:** Xi Xiong, David Tak Wai Lui, Carlos King Ho Wong, Ian Chi Kei Wong.

**Methodology:** Xi Xiong, David Tak Wai Lui, Carlos King Ho Wong.

**Project administration:** Carlos King Ho Wong, Ian Chi Kei Wong.

**Resources:** Carlos King Ho Wong, Ian Chi Kei Wong.

**Supervision:** Carlos King Ho Wong, Ian Chi Kei Wong.

**Validation:** Matthew Shing Hin Chung, Ivan Chi Ho Au.

**Visualization:** Xi Xiong, Matthew Shing Hin Chung, Ivan Chi Ho Au.

**Writing – original draft:** Xi Xiong, David Tak Wai Lui, Carlos King Ho Wong.

**Writing – review & editing:** David Tak Wai Lui, Francisco Tsz Tsun Lai, Eric Yuk Fai Wan, Celine Sze Ling Chui, Xue Li, Franco Wing Tak Cheng, Ching-Lung Cheung, Esther Wai Yin Chan, Chi Ho Lee, Yu Cho Woo, Kathryn Choon Beng Tan, Carlos King Ho Wong, Ian Chi Kei Wong.

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
