## [Editor Report · Decision Letter 0]

28 Feb 2023

Dear Dr Wong, 

Thank you for submitting your manuscript entitled "Incidence of diabetes following COVID-19 vaccination and SARS-CoV-2 infection: a population-based cohort study" for consideration by PLOS Medicine.

Your manuscript has now been evaluated by the PLOS Medicine editorial staff as well as by an academic editor with relevant expertise and I am writing to let you know that we would like to send your submission out for external peer review.

Please re-submit your manuscript within two working days, i.e. by Mar 02 2023 11:59PM.

Kind regards,

Callam Davidson

Senior Editor

PLOS Medicine

---

## [Decision Letter · Decision Letter 1]

30 Mar 2023

Dear Dr. Wong,

Thank you very much for submitting your manuscript "Incidence of diabetes following COVID-19 vaccination and SARS-CoV-2 infection: a population-based cohort study" (PMEDICINE-D-23-00434R1) for consideration at PLOS Medicine. 

[LINK]

In light of these reviews, I am afraid that we will not be able to accept the manuscript for publication in the journal in its current form, but we would like to consider a revised version that addresses the reviewers' and editors' comments. Obviously we cannot make any decision about publication until we have seen the revised manuscript and your response, and we plan to seek re-review by one or more of the reviewers. 

We hope to receive your revised manuscript by Apr 20 2023 11:59PM. Please email us (plosmedicine@plos.org) if you have any questions or concerns.

We look forward to receiving your revised manuscript. 

Sincerely,

Callam Davidson, 

PLOS Medicine

plosmedicine.org

Abstract Methods and Findings:

* Please include the important dependent variables that are adjusted for in the analyses.

Citations should be in square brackets, and preceding punctuation.

Please add the following statement, or similar, to the Methods: "This study is reported as per the Strengthening the Reporting of Observational Studies in Epidemiology (STROBE) guideline (S1 Checklist)."

When completing the checklist, please use section headings and paragraph numbers, rather than page numbers.

Did your study have a prospective protocol or analysis plan? Please state this (either way) early in the Methods section.

Lines 243-245: Please remove the sentence relating to data availability as this information is already captured via the Submission Form questionnaire. Please note that a study author cannot be the primary contact for data requests. 

Line 310: Please avoid statements of primacy and include ‘to our knowledge’, or similar. 

Lines 443-445: Please remove the lines relating to funding from the Acknowledgements (this already appears in the Financial Disclosure).

Please ensure that any references to online-only sources include a date of accession. 

Comments from the reviewers:

Reviewer #1: 

This study evaluates the risk of a new diabetes diagnosis after Covid infection and Covid vaccination. The study draws on electronic health records data for a large population in Hong Kong. Through comparison with propensity score matched comparators, the study finds that Covid infection is associated with increased diabetes incidence, but Covid vaccination is not. The study suggests that diabetes risk is lower with the Omicron variant, and that vaccination status may influence risk of Covid-associated diabetes.

The findings with respect to Covid-19 infection and diabetes are confirmatory. However, the findings with respect to vaccination and vaccination status are more novel. The study appears to be generally well conducted. However, the clarity of the paper could be improved. Multiple different hypotheses are tested but the plan of analysis is not clear. Often, approaches with respect to different hypotheses are expressed in the same sentence. 

Furthermore, the vaccination cohort was contaminated with Covid-19 infection, and the Covid-19 cohort by vaccination. The analytical approach may be pragmatic but these results are not fully transparent. 

1. At line 134, please give more details on the provenance of the two vaccines. Also, in terms of context, please add in which jurisdictions the vaccines are in current use.

2. Please also explain the vaccination policy. How were patients allocated to CoronaVac or BNT? Was the policy for two doses of the same vaccine or one dose of each? (as noted at line 221).

3. For context, please mention the vacccine uptake rates and associations with sociodemographic variables.

4. At line 168, those who did not receive any vaccines were controls presumably and not part of the vaccine cohort, it is not made clear.

5. The statistical analysis section could be clearer and and analysis plan should be provided as supplementary material. There are several hypotheses: that vaccination is associated with diabetes; that the two vaccines differ; that Covid-19 is associated with diabetes; that the risk varies for different variants; that the effect of Covid 19 is modified by vaccination etc. It should be explained how each hypothesis is tested. How will type 1 error rate be controlled? 

It is unclear, for example, how propensity score matching was used. Controls, it seems, were matched on age and sex only? (lines 259 to 263 refer to 'controls' and 'matched controls'; lines 274-5 refer to 'matched controls' and 'matched unvaccinated people').

6. Choice of the Cox model may not be optimal since it is likely that any increase in risk is transient after vaccination or infection events, whereas the hazard ratio represents an average over the time of study.

7. How will type 1 error be controlled, and rather than referring to P<0.05 as significant, please use the ASA guidelines on P values as quantifying the strength of evidence.

https://www.tandfonline.com/doi/full/10.1080/00031305.2016.1154108

8. Line 268, state how many cases of diabetes were identified.

9. Looking at Table 1, it may appear surprising that half of all participants had pre-diabetes as noted in the discussion.

10. Also in Table 1, it may not be clear why 0% of the vaccination cohort are fully vaccinated but more than 40% of the Covid-19 patients are fully vaccinated. (add foot note)

11. Table 2, the 'Rate' and 'Estimate' are ambiguously labelled and the 'rate' could perhaps be omitted.

12. Table 3 may be better presented as a Forest plot. Type 1 diabetes data is uninformative.

Reviewer #2: Alex McConnachie, Statistical Review

The paper by Xi Xiong and colleagues examines the incidence of diabetes following COVID-19 vaccination or infection, using national public health datasets from Hong Kong. This review considers the use of statistics in the paper.

In general, the statistical methods look good.

The exposed and unexposed cohorts are clearly defined. Propensity score matching is used to control for confounding due to measurable differences between the exposed and unexposed cohorts. The matching procedure looks good, though I have read some criticisms of the nearest neighbour within caliper method; did the authors consider any alternative approaches to using the propensity scores (as a sensitivity analysis, perhaps), such as inverse probability weighting, or an alternative matching procedure? 

Cox models are used to estimate associations. It is not clear whether and how the analysis takes account of the pair-matched structure in the data. Also, were the proportional hazards assumptions checked for these models? These points could be clarified.

Subgroup analyses are performed, and interaction p-values are reported (in the tables) for these, which is good. When reported in the text, only the lack of associations between infection and diabetes in younger people, and in those fully vaccinated, are reported - it is not made clear that these represent significant interactions, and that the associations are stronger in older, and particularly unvaccinated individuals. Also, the interaction with prediabetes is not mentioned - it is only stated that the results are consistent with the overall results, despite an apparently stronger association in those without diabetes. Admittedly, this may be something to do with the inclusion criteria that required an HbA1c measurement.

As an aside, Table 3 would be easier to read if there were a slight separation, or horizontal lines, between collections of subgroups. Only the overall section of this table needs to be reported. The Type I section shows virtually no data, and the Type II data are almost identical to the overall figures.

The authors show that the PS-matching procedure produces well-matched groups, by showing the data and the standardised mean differences. However, they do not report the characteristics of the exposed and unexposed groups prior to matching, except for a few words in the text of the paper, starting at line 256; Table 1 only shows summary data after matching. Perhaps a table could be added to the supplement, showing the full data, and the SMDs, prior to matching?

Saying that, in Table 1, the mean ages for the vaccinated groups appear to be about 2 years less than for their matched controls - is this true, or an error in the tables?

Reviewer #3: There is still a lack of studies on whether the COVID-19 vaccination has an impact on the incidence of diabetes.

This is a well designed, very large study from Hong Kong which shows no increased risk of diabetes after COVID vaccination. Moreover, this study suggests that the diabetes risk after SARS-CoV2 infection is not increased in fully vaccinated persons, and also depends on the type of Corona virus.

I have a few comments on the paper:

1. The following publication should also be considered in the discussion: 

Kwan AC, Ebinger JE, Botting P, Navarrette J, Claggett B, Cheng S. Association of COVID-19 vaccination with risk for incident diabetes after COVID-19 infection. JAMA Netw Open 2023; 6(2):e2255965. doi:10.1001/jamanetworkopen.2022.55965.

2. For BNT162b2, the incidence rate is actually lower for vaccinated than for non-vaccinated persons (cf. that lower risk was also observed by Kwan et al. in the aforementioned paper). The authors do not discuss this but simply state that the incidence is not larger. If you do not think that the decreased risk is causal, what are the reasons for this view?

3. For the control groups, a pseudo-index was assigned. Please give a short explanation in the methods section how this was done. 

Minor comments:

4. Page 5, line 96 / 98: Please add the time periods to which the excess risks refer to.

5. Page 9, line 195: Why do the authors not use ICD-10?

Reviewer #4: The authors used two large cohorts identified by electronic health databases in Hong Kong to evaluate the incidence of diabetes following COVID-19 vaccines and after SARS-CoV-2 infection. For both cohorts, controls were 1:1 matched using propensity scores and HRs for incident diabetes were estimated using Cox regression models. It was found that COVID-19 vaccination was not associated with an increased risk of incident diabetes, while the risk of incident diabetes increased following SARS-CoV-2 infection. This study makes a useful contribution to the literature regarding the incidence of diabetes following COVID-19 vaccination and SARS-CoV-2 infection among the Asian population. I have two major comments which I hope can help the authors further improve the work and its dissemination.

1. It will be useful to have a paragraph to briefly introduce the COVID-19 pandemic and key policies in Hong Kong, e.g. when did the first patient appear in Hong Kong, when did Hong Kong open up and what's the case number before and after that, what's the vaccination policy (who were prioritized), etc. Such information will help the audience better understand the study design and the interpretation of the results. 

2. The authors mentioned in the method part that previous SARS-CoV-2 infection was adjusted as a binary variable in the vaccination cohort. How many people had previous SARS-CoV-2 infection in the vaccination cohort? There might be an interaction effect between previous infection and vaccination on the risk of incident diabetes, so it's probably better to drop people who had previous infections in this cohort or conduct a stratification analysis.

Reviewer #5: It is a fundamental and quite impressive study assessing the incidence of diabetes after COVID vaccination based on large and representative cohort in HK. I have several comments.

1.Please describe/estimate the prevalence of using the two kinds of vaccinations in HK.

2. An individual could belong to both the two cohorts.e.g., one could get vaccination but still be infected. Although you conducted sensitivity analysis to adjust the potential impact, do you also consider to conduct subgroup analysis for the persons who were vaccinated yet infected ?

3.Do you consider those diagnosed as Gestational diabetes? If not, how to separate it from the type I and type II diabetes?

4. Line 275-280. BNT162b2 recipients were associated with lower risk, please state this point. 

5. It seems that the authors did not collect info on diabetic complications, which should be mentioned as a limitation.

[LINK]

---

## [Decision Letter · Decision Letter 2]

26 May 2023

Dear Dr. Wong,

Thank you very much for re-submitting your manuscript "Incidence of diabetes following COVID-19 vaccination and SARS-CoV-2 infection: a population-based cohort study" (PMEDICINE-D-23-00434R2) for review by PLOS Medicine.

I have discussed the paper with my colleagues and the academic editor and it was also seen again by xxx reviewers. I am pleased to say that provided the remaining editorial and production issues are dealt with we are planning to accept the paper for publication in the journal.

[LINK]

We look forward to receiving the revised manuscript by Jun 02 2023 11:59PM.   

Sincerely,

Katrien Janin, PhD

Senior Editor 

PLOS Medicine

plosmedicine.org

Requests from Editors:

AUTHORS SUMMARY

In the final bullet point of ‘What Do These Findings Mean?’, please include the main limitations of the study in non-technical language.

REFERENCES

For references to online-only sources, please change [cited 2023 April 06] to [accessed on 2023 April 06].

Comments from Reviewers:

Reviewer #1: The authors have given a comprehensive and thoughtful response to the reviewer comments, with a much improved paper.

Minor comments:

1. The Abstract could be more concise overall. However, the number of diabetes cases in each cohort should be mentioned. Absolute as well as relative risks should be given for the vaccine cohort (as for the Covid 19). It might be useful to estimate the nuber needed to harm in terms of number of Covid infections for one additional diabetes case.

2. In Table 1, the % signs could be omitted as the column headings indicates these are %.

3. In Table 2, the column heading should say OR not AND where it refers to vaccine recipients and Covid 19 cases. It may not be necessary or helpful to present the 'cumulative incidence rate' data, as this does not contribute towards interpretation. It might seem confusing that the Covid-19 cases have lower cumulative incidence rates than either vaccination group, but this is explained presumably by the shorter follow-up time. Therefore, it might be more useful to present person time in this column, to make this more transparent, and omit the cumulative incidence rates. For consistency then, the crude incidence rates will be the metric to present in the Abstract.

4. In the Abstract and Discussion, where it says 'We showed that neither BNT162b2 nor CoronaVac was associated with increased risks of incident diabetes following COVID-19 vaccination.' it would be better to say 'There was no evidence of increased risks of incident diabetes following COVID-19 vaccination' or similar. We cannot say there is no association, rather we found no evidence of an association. 

Reviewer #2: Alex McConnachie, Statistical Review

I thank the authors for their consideration of my original comments, and I am happy with their responses.

I have only one additional comment to make. It occurs to me that those with COVID-19 infection are likely to have more interaction with health services as a result of their symptoms. During this time, they may be more likely to undergo tests which might lead to an incidental diagnosis of diabetes. Those without COVID-19 infection will have less contact with health services, and therefore fewer diagnostic tests, and hence a lower chance of diabetes being detected. This could explain the association observed.

I believe this fits with Figure S4, which seems to show that the increased risk of diabetes is limited to the first few weeks after infection, with the incidence running roughly parallel after that.

This interpretation also fits with the lower increased risk following omicron infection, since (I believe) infection with the omicron variant is associated with less severe symptoms. Therefore, whilst interaction with health services will be increased in those with symptoms, the effect will be less than with other variants.

Similarly, vaccination reduces the severity of COVID-19 infections, so whilst those who were vaccinated still have an increased chance of being diagnosed after COVID-19 infection, the increase is less than in unvaccinated people.

I believe this possible interpretation should be mentioned. Apologies if it is, and I missed it in the paper.

Reviewer #3: The authors addressed all my comments and suggestions, and I recommend their paper for publication.

Reviewer #5: The revised paper address all my concerns and thus could be accepted.

[LINK]

---

## [Decision Letter · Decision Letter 3]

7 Jul 2023

Dear Dr Wong, 

On behalf of my colleagues and the Academic Editor, Amitabh Bipin Suthar, I am pleased to inform you that we have agreed to publish your manuscript "Incidence of diabetes following COVID-19 vaccination and SARS-CoV-2 infection: a population-based cohort study" (PMEDICINE-D-23-00434R3) in PLOS Medicine.

Before your manuscript can be formally accepted you will need to complete some formatting changes, which you will receive in a follow up email. In additional to formatting changes, we have one small editorial request: please add 'Hong Kong' or similar in the title.

Please be aware that it may take several days for you to receive this email; during this time no action is required by you. Once you have received these formatting requests, please note that your manuscript will not be scheduled for publication until you have made the required changes.

PRESS

Sincerely, 

Katrien Janin, PhD 

Senior Editor 

PLOS Medicine